# Components of Perinatal Palliative Care: An Integrative Review

**DOI:** 10.3390/children10030482

**Published:** 2023-03-01

**Authors:** Laure Dombrecht, Kenneth Chambaere, Kim Beernaert, Ellen Roets, Mona De Vilder De Keyser, Gaëlle De Smet, Kristien Roelens, Filip Cools

**Affiliations:** 1End-of-Life Care Research Group, Vrije Universiteit Brussel (VUB) & Ghent University, 1090 Brussels, Belgium; 2Department of Obstetrics, Women’s Clinic, University Hospital Ghent, 9000 Ghent, Belgium; 3Faculty of Medicine and Health Sciences, Ghent University, 9000 Ghent, Belgium; 4Department of Neonatology, Universitair Ziekenhuis Brussel, Vrije Universiteit Brussel, 1090 Brussels, Belgium

**Keywords:** perinatal palliative care, palliative care, perinatal period, end-of-life care, integrative review

## Abstract

When a severe diagnosis is made before or after birth, perinatal palliative care (PPC) can be provided to support the infant, parents and involved healthcare providers. An integrative and systematic overview of effectiveness and working components of existing PPC programs was needed. An integrative search was conducted in MEDLINE, Embase, CENTRAL, CINAHL, PsycInfo and Web of Science. Study designs examining the effect of PPC compared to regular care, and (empirical) articles describing the components of care included in existing PPC initiatives were included. Three independent authors reviewed titles, abstracts and full texts against eligibility criteria. PRISMA guidelines were followed; 21.893 records were identified; 69 publications met inclusion criteria. Twelve publications (17.4%) discussed the effect of a PPC program. Other publications concerned the description of PPC programs, most often by means of a program description (22/69; 31.9%), guidelines (14/769; 20.3%) or case study (10/69; 14.5%). Outcome measures envisioned four main target categories: care coordination, parents and family members, care for the fetus/neonate and healthcare providers. No trials exist to date. Analysis of working components revealed components related to changes directed to the policy of the hospital wards and components involving actual care being provided within the PPC program, directed to the fetus or infant, the family, involved healthcare providers or external actors. PPC is a growing research field where evidence consists mainly of descriptive studies and guidelines. The extensive list of possible PPC components can serve as a checklist for developing future initiatives worldwide. PPC includes several important actors: the fetus/infant and their family and included healthcare providers on both maternity and neonatal wards. This leads to a large variety of possible care components. However, while some studies show proof of concept, an evidence base to determine which components are actually effective is lacking.

## 1. Introduction

Despite increased medical and technical intervention techniques [1], parents and healthcare providers are still confronted with fetal and neonatal death [2,3] (i.e., from 22 weeks’ gestation until 28 days after birth [4]). When a severe diagnosis is made either before or after birth [5], perinatal palliative care can be provided as a multidisciplinary care approach aimed to improve the quality of life of patients and their families through prevention and relief of symptoms, whether physical, psychological, social or spiritual (WHO definition [6]). The emerging field of perinatal palliative care currently lacks an integrative and systematic overview of components included in perinatal palliative care programs, and therefore, it seems that a large variation amongst existing programs occurs internationally [4,7,8,9,10,11].

In contrast to palliative care research in adults, palliative care in obstetrics and neonatology is a relatively new field of care where the possible beneficial impact is hypothesized but has not yet been extensively evaluated [12]. Several reports on perinatal palliative care protocols, teams or educational interventions exist, though the reported protocols or interventions are often not evidence-based, and research on their feasibility and effects is lacking [13]. Existing single-center studies show proof of concept regarding the implementation of palliative care guidelines by showing improvements in reorientation of care from invasive treatments to comfort (palliative) care [14] and increased use of comfort medication (benzodiazepines and/or opioids) [15] reducing unnecessary suffering of hospitalized neonates who are dying. Furthermore, components mentioned in existing perinatal palliative care programs seem to show a large variation without insight into the outcomes or effect evaluation of each provided component [4,7,8,9,10,11]. For example, additional to providing comfort care for the child, perinatal palliative care could address the complex family needs in an emotionally turbulent time by providing a family-centered approach with a focus on parental (spiritual and cultural) values, memory making and compassionate communication between parents and providers [4,16,17]. Lastly, existing programs offering palliative care in the perinatal period often focus solely on either neonatal palliative care [18,19] or the relatively small group of families who receive a severe fetal diagnosis before birth and decide to continue the pregnancy [20,21]. This can lead to families being excluded from receiving appropriate palliative care, such as families opting to terminate the pregnancy after a severe fetal diagnosis, and failing to provide continuity of care for families transitioning from prenatal to neonatal care teams.

We therefore need to systematically review the literature describing existing perinatal palliative care initiatives internationally and the extent to which they have been evaluated so that relevant working components can be summarized and used as a basis to develop an evidence-based perinatal palliative care intervention. Therefore, our objective was to conduct a comprehensive integrative literature review, focusing on:
The outcome and effectiveness evaluation of perinatal palliative care compared to regular care provided on:➢Pain and/or symptom relief;➢Quality of care;➢Quality of life of fetuses/infants, parents, family members and involved healthcare providers;
Detailing which working components are incorporated in such perinatal palliative care programs.

## 2. Materials and Methods

We conducted an integrative review of the published literature.

### 2.1. Registration

The study protocol was registered through the PROSPERO registry of systematic reviews (CRD42021239005). The Preferred Reporting Items for Systematic Reviews and Meta-Analyses (PRISMA), a checklist used to transparently report on methodology and findings for systematic reviews, was followed in this article, see Appendix A.

### 2.2. Search Strategy

The integrative search was conducted in MEDLINE, Embase, CENTRAL, CINAHL, PsycInfo and Web of Science. A search was performed on 23 February 2021, and an update was conducted on 31 May 2022. Language was not limited; the time of publication was limited to 1 January 1997 and onwards because the first commentary on perinatal palliative care was published in 1997 [22], which was later identified as the start of perinatal palliative care [23]. The MEDLINE syntax was developed alongside information specialists and used as a starting point for the construction and validation of the search strategy, hereby adhering to the Peer Review of Electronic Search Strategies (PRESS) criteria for electronic search strategies [24]. The MEDLINE search string can be found in Table 1; a detailed search strategy is provided in Appendix A.

### 2.3. Article Eligibility Criteria

#### 2.3.1. Article Designs

We included all empirical studies that examine the effect of perinatal palliative care on pain and/or symptom relief, quality of care for, and well-being of fetuses/infants, parents, family members and involved healthcare providers. This includes interventional, observational and survey designs (pre-post study designs, non-randomized trials, randomized controlled trials, cohorts, case-control designs, cross-sectional designs, prospective survey designs and retrospective survey designs) and qualitative studies. Additionally, to examine which working components are included in existing perinatal palliative care programs, protocol papers or studies with a single case design discussing the development, description and/or evaluation of such programs were included.

#### 2.3.2. Population

All publications that concerned palliative care for fetuses or neonates in the perinatal period, their parents and involved healthcare providers and met with the article design inclusion criteria mentioned above, were included. The study population was defined as fetuses or neonates from 22 weeks of gestation up until 28 days after birth with a serious illness where palliative care is being provided. Articles solely focusing on pediatric palliative care for infants being cared for in a pediatric ward and/or older than the age of one month were excluded, as well as articles solely focusing on adult palliative care (18+). For the used definition of perinatal palliative care, see Table 2.

#### 2.3.3. Intervention

All publications meeting the following criteria were included: (a) They either evaluate the effect of or describe a program, guideline or protocol concerning perinatal palliative care, which constitutes a multi-component program (see definition provided in Table 2); (b) the program, guideline or protocol concerns providing palliative care for fetuses and infants with a serious illness in the perinatal period; and (c) included programs needed to at least identify themselves as a perinatal, prenatal and/or neonatal palliative care or perinatal, prenatal and/or neonatal hospice program.

### 2.4. Article Selection

All records were exported to the reference management software EndNote (Version X9). Duplicated records were removed. Using the Rayyan tool [25], three authors screened titles and abstracts: LD screened all records, and MD and GD independently screened half of all records. Any disagreement between reviewers was resolved by discussing the eligibility criteria between all three reviewers, and by verifying the full-text article if necessary. Following abstract screening, full texts were screened in a similar matter by the three reviewers. One author (LD) hand-searched the reference list of included articles.

### 2.5. Data Extraction

From each selected full text article, the following information was extracted (when applicable): title, author(s), publication year, language, journal title, country, study design, study (care) setting, aim/research question(s), targeted population (perinatal, prenatal and/or neonatal), components of care included in the perinatal, prenatal or neonatal palliative care program/intervention/guideline, start of the program (in time), number of participants, envisioned outcomes and effectiveness. If data were missing, authors were not contacted for additional information. Data extraction was conducted by three reviewers (LD, MD and GD); each reviewer extracted data from one-third of all included publications and checked a 20% sample of one of the other two reviewers.

### 2.6. Data Synthesis

We summarized results in three overview tables. Firstly, publications where the effect of a perinatal palliative care intervention was examined were grouped together and looked at separately so that outcomes or effect evaluation could be discussed. Secondly, all other publications describing a program, guideline or protocol concerning palliative care were listed. Thirdly, working components mentioned in all included articles were grouped together and categorized. Within a publication, we identified working components of perinatal palliative care as every mention of a care or service provided to fetuses, infants, parents, family members, healthcare providers or other involved actors; any change made to regular provided care to include perinatal palliative care; or any means of communication between the involved actors. Any mention of such component was listed and categorized using NVivo by sorting all mentioned care components based on what care was being provided, to whom the care component was directed and what the envisioned outcome was. The categorization was made by one researcher (LD) and thoroughly corroborated by five other researchers (KC, KB, FC, KR, ER). This characterization was represented in a schematic overview.

## 3. Results

In the following sections, we will describe the article selection process (Section 3.1) and the characteristics of the included studies (Section 3.2). Afterwards, results concerning outcome measures and effect evaluation of perinatal palliative care programs will be discussed (Section 3.3). Lastly, we will provide an overview of all working components of perinatal palliative care discussed in the articles (Section 3.4), independent on whether they are evidence-based.

### 3.1. Article Selection

A total of 21.893 records were identified in MEDLINE, Embase, CENTRAL, CINAHL, PsycInfo and Web of Science (Figure 1). After deduplication, abstract and full text selection, 69 articles met the inclusion criteria. Recurring exclusion criteria were: duplications, animal studies, no topic of palliative care, sole focus on adults or pediatric palliative care and reviews of included studies, indicating that they did not meet inclusion criteria.

### 3.2. Article Characteristics

An overview of all included articles showing their study design, country, aim/research question and reference, can be found in Table 3 (publications discussing outcomes and effect evaluation of PPC) and Table 4 (other publications discussing working components of PPC). The majority of articles were located in the US (45/69; 65.2%). Twelve publications (12/69; 17.4%) discussed the outcomes or effect evaluation of a perinatal palliative care program using quantitative methods (10/69; 14.5%) or a mix of quantitative and qualitative methods (2/69; 2.9%).

The other publications (57/69; 82.6%) concerned the description of perinatal palliative care programs by means of a program description (22/69; 31.9%), guideline or recommendation (14/69; 20.3%), case study (10/69; 14.5%), retrospective chart review (4/69; 5.8%), surveys of healthcare providers (2/69; 2.9%), expert reflections (2/69; 2.9%), semi-structured interviews with PPC members (1/69; 1.4%), or program development using expert opinion (1/69; 1.4%), the Delphi method (2/69; 2.9%) and/or literature review (2/69; 2.9%).

The majority of publications discussed palliative care provided in the perinatal period (43/69; 62.3%), 24 publications focused solely on care provided in the neonatal period (24/69; 34.8%), and 2 publications focused solely on care provided before the infant was born (2/69; 2.9%).

### 3.3. Outcomes and Effect Evaluation of Perinatal Palliative Care

A total of 12 out of the 69 included publications assessed the effect of perinatal palliative care (12/69; 17.4%). Study designs included (retrospective) chart review (5/69; 7.2%), prospective cohort study (2/69; 2.9%), surveys (3/69; 4.3%) and a case audit (1/69; 1.4%). In one publication, the study design was not reported (1/69; 1.4%). None evaluated the effect of a perinatal palliative care intervention using a (randomized controlled) trial. Envisioned main outcomes of discussed studies prior to implementation were not reported, and (causal) relations between perinatal palliative care components and measured outcomes were not documented. Outcome measures were divided based on four main target categories: the effect of palliative care interventions or programs on care coordination, the effect on parents and family members, the effect on care for the fetus and/or neonate and/or the effect on healthcare providers.

#### 3.3.1. Care Coordination

Seven studies measured their effect of perinatal palliative care in terms of changes to the organization and coordination of care. Bolognani et al. [26] mentioned that a perinatal palliative care program facilitated decision making and consensus building, improved the therapeutic alliance and led to a better flow of information between all involved. Similarly, by implementing a palliative care checklist, Taylor et al. [32] indicated improvements in documentation on, amongst others, comfort care, monitoring of fluids and nutrition and discussions with parents. Psychological support, however, was still rarely offered [32]. Rusalen et al. [30] indicated that despite the implementation of a perinatal palliative care protocol, the number of eligible newborns actually receiving perinatal palliative care was still limited [30]. The neonatal palliative care intervention of Samsel and colleagues [14] showed an increase of redirection of care from curative to palliative over the course of the intervention, but contact with social work and chaplaincy did not significantly increase. Tewani and colleagues [33] reported that in a new perinatal palliative care service in Singapore, care plans including resuscitation and other decisions were made before birth in 63% of cases. Furthermore, 44% of cases died during the hospital stay, and of those discharged home, 24% were still being supported by the community palliative team [33]. Lastly, Younge et al. [15] indicated that implementation of a neonatal palliative care program in the neonatal intensive care unit caused a rise in the number of meetings to discuss end-of-life care and an increase in time between the first end-of-life discussion and withdrawal of life support. Clinical factors, such as mortality, postnatal age at death, withdrawal of life-support, incidence of DNRs, amount of morphine administration in the last 24 h and the number of infants receiving neuromuscular blockers, however, did not differ before or after implementation of the program [15].

#### 3.3.2. Parents and Family Members

In 5 studies (5/12; 41.7%) discussing the effect of perinatal palliative care, the measured outcomes were targeted towards the parents and family. Petteys and colleagues [19] indicated in their prospective cohort study that provision of neonatal palliative care did not increase stress in parents and may even decrease stress in parents of the frailest and/or sickest infants. Furthermore, in another prospective cohort study by Callahan et al. [27], perinatal palliative care even reduced parental stress in parents of neonates with congenital heart disease, though parental depression and anxiety did not decrease. Survey studies indicated that parents who received palliative care for their fetus and/or infant showed to be highly [29] to even extremely [19] satisfied with the care they received. Palliative care was seen to increase feelings of understanding the diagnosis [29], and the service provided was seen as valuable and helpful [29,31]. Lastly, the retrospective chart review of Jalowska and colleagues [28] showed that all mothers in the newly set up perinatal hospice program wanted to see and hug their child and wanted to participate in memory making, which was expressed to be extremely important.

#### 3.3.3. Fetus and/or Neonate

Some 33.3% of studies (4/12) discussing an effect of perinatal palliative care focused on the effect on the fetus, infant or neonate. Using a survey design, Parravicini et al. [18] reported that parents felt that their babies were kept comfortable and were treated with respect and compassion after a standardized neonatal comfort care program was implemented. Additionally, both Younge et al. [15] and Samsel et al. [14] used retrospective analysis of charts to indicate that implementation of a palliative care intervention ensured management of pain control in neonates during the final days. Taylor and colleagues [32] analyzing case audits, however, did not find that implementation of a neonatal palliative care intervention caused significant differences in pain alleviation.

#### 3.3.4. Healthcare Providers

Only 1 publication (1/12; 8.3%) focused on the effect of perinatal palliative care on healthcare providers. Steen [31] indicated that implementing a perinatal bereavement care program led to increased confidence of healthcare providers in caring for families due to the education, support and mentoring they received.

### 3.4. Working Components of Perinatal Palliative Care

All discussed working components of perinatal palliative care programs in the 69 included publications were extracted (see Figure 2 and Table 5). Ten main components and their subcomponents focused on changes directed to the policy of the hospital wards, thereby making changes to routines and daily practice of involved healthcare providers. Six main components involved care directed to the individual recipients of the perinatal palliative care program and can thus be considered as care components. References of all components to each included publication can be found in Table 5.

#### 3.4.1. Practical Organization of a Perinatal Palliative Care Team—10 Main Components

*Perinatal palliative care team members*: Firstly, members of the core perinatal palliative care team need to be selected. Variation existed among programs included in this review; however, physicians (either an OB-GYN, a neonatologist and/or a pediatrician were most common), nurses and/or midwives, and psychosocial support such as a social worker and/or a psychologist were generally considered part of the core team. Furthermore, a care coordinator was often included to guide family members through the process from diagnosis until death/stillbirth and bereavement, their main goal consisting of being a familiar face and keeping up with scheduled appointments. Additionally, team members could be added when needed for a specific case, including organ specialists, religious and/or spiritual support providers, general practitioners, etc.*Activation of the perinatal palliative care team*: When providing perinatal palliative care, decisions need to be made concerning who is eligible to receive such care and when the perinatal palliative care team will be involved. Eligibility criteria varied from severe prenatal diagnosis to (extremely) premature infants and neonates with serious illness diagnosed after birth. Furthermore, some programs indicated their involvement solely when the pregnancy of a child with a severe prenatal diagnosis would be carried to term, whilst a limited number of programs included care for families opting for a (late) termination of pregnancy. Lastly, programs could be included in care whilst curative treatment was still sought, while others were only called when curative care was deemed futile.*Providing a place and/or circumstances for palliative care*: A maternity ward or neonatal intensive care unit is often not equipped as an ideal environment for families saying goodbye to their child. Providing adequate perinatal palliative care thus often included adjustments to the environment to ensure privacy for the grieving family and peace and quiet from the loud intensive care unit to enjoy limited quality time together. It also means to allow bonding with others, including (grand)parents, siblings, relatives and friends.*Step-by-step plan:* A checklist or protocol was often included in the perinatal palliative care program to direct involvement of the perinatal palliative care team in a structured manner.*Organization of communication between the care team*: A clear and detailed way of documenting is needed to record all conversations and decisions made for a child and/or family. Electronic patient records could be modified or separate shared documents could be made so that information is easily shared with all stakeholders. Additionally, regular meetings with the core perinatal palliative care teams are needed to keep everyone up to date and to ensure continuity of care when transfer from the prenatal to the neonatal team is necessary.*Teaching of staff members*: Perinatal palliative care programs often provided their members, and sometimes even all members of the regular prenatal or neonatal team, with either formal training (including palliative care training, communication training, organ procurement training, etc.) or on-the-job training moments. On-the-job training moments could include, for example, regular mortality and morbidity meetings, palliative care orientation for new staff members or supervision by more experienced team members.*Collaboration with hospice or palliative home care services*: When possible, some perinatal palliative care teams set up a regular collaboration with existing (pediatric) hospice services or palliative home care services, allowing families to take their child home.*A regular audit of the perinatal palliative care approach*: Some perinatal palliative care programs specifically include fixed moments to review or update their care approach according to the newest developments and insights.*Fundraising for the program*: Several programs mentioned the need for fundraising activities to provide perinatal palliative care freely to all families in need.*Community awareness and involvement*: Informing the public on how they can aid families going through perinatal loss can add an additional layer of support.

#### 3.4.2. Care Components of the Perinatal Palliative Care Program—6 Main Components

Six main components were found on the care directed to the individual recipients of the perinatal palliative care program. Variation occurred on the number of components included in each publication, ranging between 5 and 39 (sub) components mentioned. The mean number of (sub) components mentioned per publication was 18.9 (SD = 8.3). Components could only be included in the analysis if they were explicitly mentioned in either the publication or appendix.
*Child-directed care*: These components include (medical) care provided directly to the child or fetus either before or after death. *Ensuring comfort of the child (non-medical)*: A main goal of perinatal palliative care was often to ensure that the child is comfortable by limiting unnecessary and often painful assessments and treatments, providing a stress-free environment with minimal light and/or noise, providing skin-to-skin contact, and co-bedding with multiples.*Adequate pain/symptom management*: Aside from non-medical procedures to ensure comfort, adequate pain and/or symptom management is provided to relieve pain and suffering. Additionally, validated pain and comfort scales can be used to assess pain or suffering in nonverbal neonates so that medication can be adjusted accordingly.*Food/nutrition provision*: Whilst breastfeeding when the child is able is universally considered beneficial during perinatal palliative care, variability consists in whether or not artificial food or nutrition should be provided.*Other care provided to the child*: Perinatal palliative care programs sometimes discuss other care provided to the children in the (neonatal) intensive care unit, such as dressing the child, bathing, and monitoring of vital signs, which can provide comfort to family members or healthcare providers or allow for bonding moments.*Care of the child after death*: Providing perinatal palliative care often included caring for the body after the child passed away. This encompasses bringing the body to a morgue, providing visiting options after death and options to cool and sometimes even rewarm the body.*Postmortem medical procedures*: Death of a fetus and/or neonate brings forward specific medical challenges, such as scheduling an autopsy or performing postmortem tests to confirm a diagnosis, which can be very important for family members in light of future pregnancies. Furthermore, the perinatal palliative care team often prepares for parents wanting to donate their child’s organs.*Family-directed care*: Perinatal palliative care was often discussed to be directed mainly towards family members, such as parents, siblings, grandparents and other close relatives/friends. The following care components were mentioned:*Family-centered care*: Adept care was tailored to the values, hopes, needs and cultural background of the family (within reasonable limits). Care was customized for individual family members as much as possible, including, for example, providing a translator or speaking to religious representatives of the family. Care can be tailored to the parents and family as a whole or even separately for individual family members when needed.*Promote family bonding and parenting*: Perinatal palliative care includes promoting family bonding and making memories, both during life and after death of the infant. This means supporting families to make the most out of the little time they have with their infant, including taking part in caring for their child. However, some programs mention having respect for the families’ wishes not to bond.*Family-centered psychosocial support*: Programs providing perinatal palliative care often include a strong and constantly available psychosocial support system directed towards the family, including, but not limited to, social services, social workers, psychologists, chaplains and references to support groups. Some programs pay specific attention to siblings or parents other than the mother or to providing support during a new pregnancy.*Bereavement support* and follow-up appointments with physicians, nurses, midwives, psychosocial workers or other members of the (core) team are mentioned as another essential component of family-centered psychosocial support.*Religious and/or spiritual support* can be provided if parents so wish, allowing for specific rituals and ceremonies tailored to their specific needs.*Practical support*: Families might need more practical forms of support such as aid in funeral planning, registering the birth and the death of the child and informing other organizations on the death of their child, which is also included in several perinatal palliative care programs.*Healthcare provider directed care*: A third recipient of perinatal palliative care support mentioned are the healthcare providers responsible for caring for the fetus/infant and the family. The following components are included:*Support for healthcare providers*: Some perinatal palliative care programs discuss formal or informal means of support for healthcare providers, ranging from psychological support during working hours to being able to ask questions to more experienced peers and colleagues.*Debriefings after death*: Formal debriefings of the multidisciplinary staff members after the death of a child can provide time to reflect on what happened and what could be improved. Additionally, possible conflicts, problems and issues can be addressed.*Relieve healthcare providers of other tasks when caring for an infant in their final moments*: Fixed staff members, such as nurses or physicians, are indicated to be available at all times during the dying process. Other tasks can be taken care of by colleagues during this time spent in close contact with the family.*Advance care planning*: Aside from care directed to a specific recipient of perinatal palliative care, a large care component focusses on advance care planning when the medical situation allows it. This means that time is available to make decisions beforehand instead of during crisis situations. Different scenarios of care are discussed with healthcare providers and parents/family members so that a future treatment plan can be set up. We identified the following subcomponents:*Care plan during pregnancy and/or life of the child*: When a prenatal diagnosis is made, the prenatal care plan can be discussed beforehand, including decisions on whether or not all regular examinations are still needed or wanted during pregnancy and how maternal health, both psychosocial and medical, will be assessed. An important prenatal decision in case of severe fetal diagnoses is whether curative treatment, non-aggressive obstetric management, neonatal palliative care or (late) termination of pregnancy with or without feticide is preferred. Furthermore, a birth plan can be made, including deciding on the mode and time of delivery, who should be present, how maternal comfort will be assured and whether fetal monitoring is required. Lastly, a neonatal care plan can be made when survival past birth is possible or when the severe diagnosis is made post birth. The neonatal care plan includes discussing curative and palliative care (both can occur simultaneously); discussing possible courses of action depending on the medical situation of the child, such as resuscitation, treatments and possibly limiting care, and confirming the prenatal diagnosis neonatally.*Death plan*: When death or stillbirth is a possibility, healthcare providers can provide parents/family members with possible scenarios of what could happen. These conversations include inquiring about preferences regarding religious ceremonies, washing and clothing the infant, taking pictures and making memories. Additionally, a bereavement care plan can be set up, where healthcare providers and parents discuss when follow-ups will take place and what support parents anticipate needing.*Regular revisions of the care plan*: Preferences of the parents (and the healthcare providers) can change rapidly when new information becomes available or when individuals change their minds. The care plan can thus be reassessed regularly and is considered flexible.*Components regarding the decision-making process with parents/family members:* When a severe diagnosis is made before or after birth, decisions can often be made beforehand. Here, advance care planning can take place, yet this is not always the case. Irrespective of when these decisions are made, key care components in perinatal palliative care focus specifically on which information is shared, how this information is shared, who makes decisions and how conflicts are handled. *Which information is shared*: Information regarding diagnosis and prognosis is shared openly and honestly, not only with parents but with the entire medical team including nurses, psychologists and social workers. Prognostic uncertainty can be addressed, and parents need to be prepared for possible disease trajectories. Healthcare providers sometimes raise topics for discussion in family meetings when parents leave them open, such as discussing what they can expect during the dying phase. All possible treatment options and their consequences are provided, including curative options, limiting treatment, providing comfort care or termination of the pregnancy.*The manner of sharing information*: Regular, planned and formal consultations with parents are to be scheduled. During these consultations, parents have the ability to see all members of the care team so that they are adequately informed. A care coordinator can be useful as a set point of contact for parents so that someone is (almost) always available for questions. The provided information is presented in an understandable way to parents and is thus often tailored to their knowledge level, norms and (cultural) values. Translators are involved when needed, and information is repeated multiple times. Comprehension is often assessed. Information is provided in a compassionate manner, respecting norms, values and wishes. If conflicts arise concerning these norms, values and wishes, conflict resolution might be needed (see further).*Shared decision making*: Inquire about the role parents would like to play in decision making. Shared decision making is recommended in most programs; however, some parents indicate not wanting to be involved. Healthcare providers support patient (and in this case parent) advocacy, empowering parents to participate in decision making.*Conflict resolution*: Conflicts between healthcare providers and parents can arise concerning, for example, treatment options, limiting care, involvement in decision making, and openly sharing information. An ethics consult or (cultural) mediation is sometimes provided, and parents are provided options to seek a second opinion. Additionally, for conflicts arising between healthcare providers of the team, perinatal palliative care programs include the possibility to step down from a case, providing mediators and accurately debriefing conflicts afterwards.
*Care of externals:* Four perinatal palliative care programs briefly mentioned support for externals. This component differs significantly from the previous care components, which are directed towards the child/fetus or actors in direct contact with a specific child/fetus. This component includes:*Care for other families at the ward*: In the neonatal ward, families often share rooms. The death of a child in the ward can take away hope for their own child, which was in this case adequately addressed to other families.


## 4. Discussion

Using an integrative review design, we identified 69 publications that reported a perinatal palliative care program. Only twelve of those discussed the effect of such palliative care programs, thereby mainly discussing what changed in the daily provision of care, and care given to infants, parents and family members and involved healthcare providers. The limited evidence relating to outcomes and effect evaluation mostly provided proof of concept of perinatal palliative care, showing that organizing such care within regular care provided is feasible. We found a large variety in the amount of care components included in perinatal palliative care programs. Many focused on organizational changes needed to accommodate such programs; care provided to fetuses/infants, parents and family members involved healthcare providers and external actors, advanced care planning and the decision-making process.

### 4.1. Considerations

We used broad inclusion criteria and a broad search string to identify the literature on perinatal palliative care programs, leading to a lack of specificity and a large amount of irrelevant articles which needed to be filtered out during the title and abstract selection phase. As perinatal palliative care is a new and emerging field [12], variation in terminology exists; thus, the broad search string was needed to ensure that no relevant information was missed.

Two out of three included articles discussed perinatal palliative care programs located in the United States, which could skew the validity of the results in favor of their care system (ex. resource funding for medical care) and potentially compromise applicability to non-US health systems.

During analysis of care components, only care components discussed in the identified publications and their Appendix A could be included. Smaller care components of a program could possibly be left out of such individual reports; however, the large number of included reports makes missing care components unlikely.

### 4.2. General Discussion

Our review of perinatal palliative care programs resulted in a multitude of care components, without an evidence base on which components have a significant effect on fetuses and infants, parents and family members, healthcare providers and others involved. No randomized controlled trials were found examining the effect of a perinatal palliative care intervention, and existing evidence is mostly based on retrospective chart reviews. A limited number of studies show proof of concept indicating that it is feasible to set up a perinatal palliative care program within standard prenatal and neonatal care. However, outcomes are often not robustly measured, and when outcomes are assessed, they mostly include evidence of a change of practice rather than examining their desired effect on, for example, patient comfort or parents’ or provider satisfaction. Outcome measures other than a change of practice mostly focus on the effect on parents and family members, measuring stress, anxiety, depression or satisfaction with care. This leaves the effects on the fetus or infant, and especially on involved healthcare providers, hardly examined. We can provide an extensive checklist on components that could be included in future perinatal palliative care initiatives, yet we can provide no insights on what actually contributes to better care. The large number of involved actors further confounds who should be the main receiver of care and what should be envisioned as the main outcome measure in future randomized controlled trials.

Providing family-directed care seemed to be the main focus of perinatal palliative care, with psychosocial and spiritual support for family members and allowing family bonding as the most frequently provided care components. Many family-directed care components, such as providing memory-making opportunities and constant availability of support, seem self-evident, yet time-constraints in a hectic intensive care unit can make providing such care difficult. Additionally, some questions and caveats remain.

Firstly, within one palliative care program, the sole recipients of care are most often live-born infants or families where a severe fetal diagnosis was made before birth where the pregnancy was continued. This could lead to fragmented care as providing care for perinatal loss is similar before and after birth [5]. Additionally, 16% of reported programs included care provision in case of termination of pregnancy. Future research should investigate whether people eventually choosing to terminate the pregnancy could also benefit from receiving perinatal palliative care so that all perinatal palliative care needs are met.Secondly, inconsistencies occur in the amount and the duration of bereavement support, ranging from follow-up phone calls [46], home visits or mail [48]; providing follow-up appointments with physicians [34] or conversations with nurses [40], psychologists, chaplains [59] or others and lasting for an undetermined [51] or fixed [60] amount of time, even years after the loss of their child [70]. This raises questions on what constitutes bereavement care and how long perinatal palliative care programs should offer this service, which is currently unclear and needs further examination.Thirdly, when providing support, the focus was most often on the mother. Only two publications explicitly mentioned support for the father or significant other [74,76], who also has the tendency to focus on the mother and her wellbeing [76]. Future perinatal palliative care programs should be aware of this and aim to provide support to both parents when applicable, tailored to their individual needs.

A second component included in all programs concerns care directed towards the neonate.

There were various ways of making sure neonates are being kept comfortable. This was performed by both non-medical treatment of the infant, such as limiting unnecessary and often painful assessments [72], skin-to-skin contact [38] and reducing noise and lights [11], as well as by medical treatment providing adequate pain and symptom relief.In all included publications, opposing views on which care should be provided was rare; however, discussion arises on whether or not artificial nutrition and hydration should be continued until death. While some programs anticipate providing parenteral or oral feeding to keep the child comfortable [55], others focused on stopping artificial nutrition and hydration during the dying process [58]. In such situations, the comfort of the child should come first; therefore, when artificial nutrition and hydration do not increase comfort, we would consider it pointless and possibly even burdensome for the child. However, breastfeeding was always considered beneficial when the child is able to [52].Lastly, care for the infant after he or she passed away seems to be exclusive to the perinatal population, including, for example, cooling and even rewarming the body [65] or procuring organs for organ donation specific to the neonatal population [11]. We feel that specialized training or information sessions for healthcare providers on these topics could be beneficial.

The main focus on parents, families and infants is understandable, yet healthcare providers seem often forgotten in guidelines and perinatal palliative care programs.

Only 22% of included programs discuss professional support for healthcare providers being tasked with providing perinatal palliative care, for example by offering consults with a psychologist, ethics consultant, psychiatry staff or social workers. Whether this lack of support is due to it being deemed unnecessary, or due to it being classed as subordinate to support for the family is unclear.Additionally, debriefings, recommended by 19% of discussed programs, could serve as a tool to reflect on what happened and address possible conflicts, problems and issues. However, no details are given on when and how often these debriefings should take place, who should participate, and what should be discussed.Lastly, while multiple programs focused on providing aid to family members during the advance care planning and decision-making process, details on supporting healthcare providers during this process were limited to recommending specialized training. We therefore feel that support for healthcare providers in perinatal palliative care could be expanded in the development of future programs and initiatives. (For more references on components raised in this discussion, see Table 5).

## 5. Conclusions

This extensive list of care components that could possibly be included in perinatal palliative care programs can serve as a checklist for developing future initiatives worldwide. Every component can be adapted to fit specific needs of the setting, hospital or country where the newly developed program would be situated. However, evidence of outcomes and effectiveness of the individual components or of perinatal palliative care programs as a whole is scarce. More research is needed to identify to what extent the various components lead to better perinatal palliative care. Parents, family members and neonates are currently the main focus of care provided in this context, yet more headway can be made in providing professional, structured and scheduled support for healthcare providers.

## Figures and Tables

**Figure 1 children-10-00482-f001:**
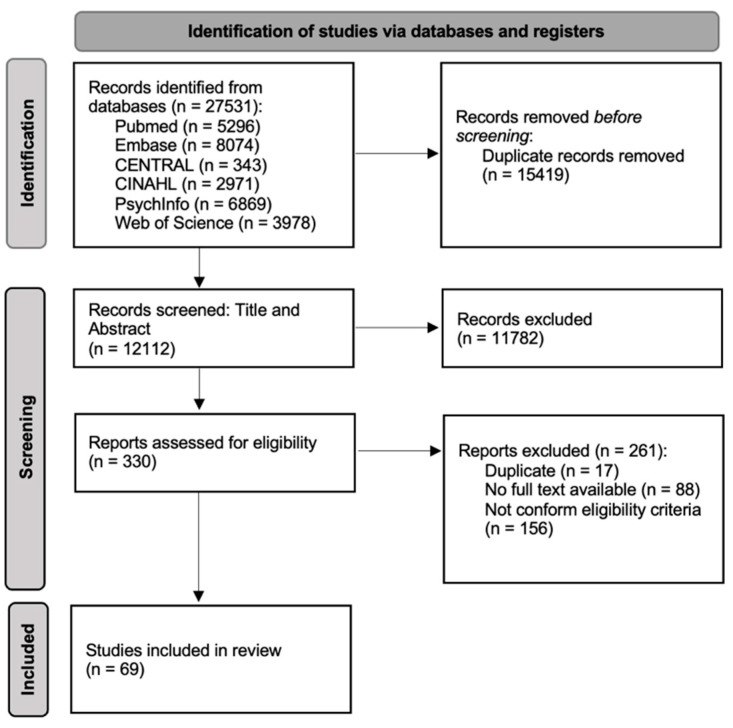
PRISMA flowchart of the article selection.

**Figure 2 children-10-00482-f002:**
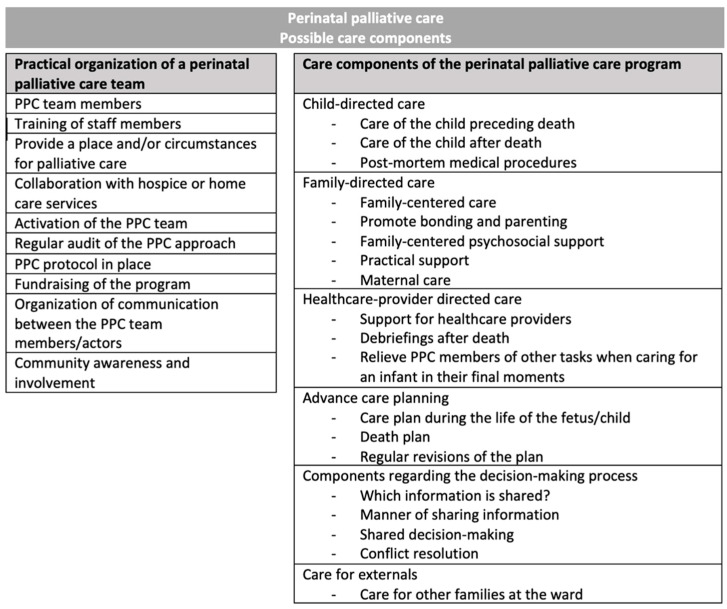
Schematic overview of all involved (care) components in PPC programs (summary, for more information, see detailed Table 5).

**Table 1 children-10-00482-t001:** Search string MEDLINE.

Search Block		Mesh	Text
Population	Fetus, neonate, infant	“infant, newborn”[MESH] “Fetus”[Mesh:NoExp]	infant* [Title/Abstract] OR newborn* [Title/Abstract] OR neonat* [Title/Abstract] OR newly born [Title/Abstract] OR newly-born [Title/Abstract] OR new-born* [Title/Abstract] Foetus [Title/Abstract] OR Fetus [Title/Abstract] OR Foetal [Title/Abstract] OR Fetal [Title/Abstract]
	Death of fetus, neonate, infant	“Infant Mortality”[Mesh] OR “Infant Death”[Mesh:NoExp] “Fetal Death”[Mesh] OR “Fetal Mortality”[Mesh] “Perinatal Mortality”[Mesh] OR “Perinatal Death”[Mesh]	(neonatal[tiab] OR perinatal[tiab] OR fetal[tiab] OR foetal[tiab] OR infant[tiab]) AND (death[tiab] OR mortality[tiab] OR demise[tiab] OR loss[tiab]) (Terminat*[Title/Abstract] AND pregnanc*[Title/Abstract])
Palliative care		“Hospice and Palliative Care Nursing”[Mesh] OR “Palliative Medicine”[Mesh] OR “Palliative Care”[Mesh] OR “Terminal Care”[Mesh]	(palliative[tiab] OR end-of-life[tiab] OR end of life[tiab] OR EOL[tiab] OR comfort[tiab] OR hospice[tiab] OR terminal[tiab]) AND (care[tiab] OR nursing[tiab] OR medicine[tiab]) palliation[tiab]
Perinatal palliative care			perinatal palliative care [Title/Abstract] OR perinatal hospice [Title/Abstract]
Not animals		NOT (“Animals”[Mesh] NOT “Humans”[Mesh])	

**Table 2 children-10-00482-t002:** Concepts and definitions.

Perinatal palliative care: For the purpose of this study, perinatal palliative care will be defined as a multi-component provision of care for fetuses or neonates with a serious illness in the perinatal period (22 weeks of gestation–28 days after birth) and their parents, families and involved healthcare providers, aimed to relieve pain and control symptoms and to improve the quality of care for, and well-being of, fetuses and infants, their families and involved healthcare providers. It is holistic, family-centered, comprehensive, and multidimensional, so that it addresses not only the physical aspect, but also the psychological, social and spiritual dimensions. A serious illness: A serious illness is defined as either a lethal diagnosis in the prenatal or neonatal period, or a diagnosis for which there is little or no prospect of long-term survival without severe morbidity or extremely poor quality of life, and for which there is no cure. We will not define a limited group of diseases for which palliative care is needed to maximize the sensitivity of our search.

**Table 3 children-10-00482-t003:** Characteristics of included publications discussing outcomes and effect evaluation of perinatal palliative care programs.

Reference	Country	Study Design	Targeted Population	Aims/Research Questions	Outcomes and Effect Evaluation
Bolognani et al. [26]	Italy	Retrospective chart review	Perinatal	To describe the model build up to take care of fetuses and newborns eligible for perinatal palliative care	Facilitated decision making and consensus building, improvement of therapeutic alliance, better flow of information.
Callahan et al. [27]	US	Prospective cohort study of parents of neonates with congenital heart disease	Neonatal	To test the hypothesis that an innovative method of early palliative care reduces psychological distress in parents of neonates with congenital heart disease	The study demonstrated that early PC reduced parental stress. Parental depression and anxiety did not decrease with the intervention.
Jalowska et al. [28]	Poland	Retrospective chart review	Perinatal	To demonstrate the role of perinatal palliative care in case of severe developmental disorder in the fetus with a potentially lethal prognosis	All mothers in the perinatal hospice program wanted to see and hug their child. All families wanted to participate in memory making and many of them expressed the importance of these memories to the team.
Loyet et al. [29]	US	Survey after start of a fetal care team	Perinatal	Improvement in quality of care for women carrying a fetus with a suspected or known fetal anomaly	Patients receiving fetal perinatal palliative care were highly satisfied and felt that support was valuable. Enhanced patients’ feelings of their knowledge of the condition or diagnosis of their fetus.
Parravicini et al. [18]	US	Prospective mixed method self-report survey	Neonatal	Assess the perception of parents concerning the state of comfort maintained in their infants affected by life-limiting conditions when treated by the neonatal comfort care program	Parents felt that their baby was comfortable and treated with respect, care and compassion by professionals. The environment was perceived as mostly peaceful, private and non-invasive.
Petteys et al. [19]	US	Prospective cohort study	Neonatal	Examine the effects of palliative care on NICU parent stress and satisfaction	PC services did not increase parent stress levels and may decrease stress in parents of the frailest, sickest infants. All PC parents were extremely satisfied with care, versus only 50% of normal care parents.
Rusalen et al. 2018 [30]	Italy	Not reported	Perinatal	To evaluate the efficacy and efficiency of their perinatal palliative care protocol	Prenatal consultations by neonatologists increased. The number of newborns who died in the NICU was reduced by 30%. The number of eligible newborns who actually received perinatal palliative care, was still limited.
Samsel et al. [14]	US	Retrospective and prospective chart review	Neonatal	Evaluating the impact of the implementation of a palliative care intervention within the NICU on outcomes of dying infants	Redirection of care increased. During the final 48 h of life, palliative medication was administered more often. Contact with social work and/or chaplaincy did not increase significantly.
Steen, S.E. [31]	US	Survey	Perinatal	To detail the specifics of a perinatal bereavement program development, and to evaluate feasibility	70% of families not only found value in having a plan, but also felt that the staff followed the proposed plan. 85% found follow-up phone call helpful, 70% found written support material helpful. Staff has increased confidence in caring for bereaved families owing to the education, support and mentoring they have received.
Taylor et al. [32]	US	Case audits on the implementation of checklists of care provided	Neonatal	To examine the quality of locally delivered neonatal palliative care before and after a regional guidance implementation	Checklist of care led to more patients being identified antenatally with life-limiting diagnoses, and timely planning for anticipated care. Improvements of documentation in domains of comfort care, monitoring, fluids and nutrition, completion of diagnostics, treatment ceiling decisions, resuscitation status and discussion with parents. Psychological support was rarely offered, most likely because of a lack of access. No significant difference in pain alleviation.
Tewani et al. [33]	Singapore	Chart review after program development	Perinatal	To develop a service providing individually tailored holistic care during pregnancy, birth, postnatal and bereavement period	Care plans including resuscitation and other decisions were made before birth for 63% of cases. Of these, 92% of parents opted for no active resuscitation. 44% demised during the hospital stay. Of those discharged home, 24% are still being supported by the community palliative team.
Younge et al. [15]	US	Retrospective study	Neonatal	To evaluate the impact of Neonatal palliative care program on end-of-life care in the NICU	Mortality, postnatal age at death, withdrawal of life-support, incidence of a DNR, amount of morphine administered in the last 24 h, and number of infants receiving neuromuscular blockers did not differ. A greater proportion of infants received benzodiazepines in the final days, the number of meetings to discuss end-of-life care was higher, time between first end-of-life meeting and withdrawal of life-support was longer.

**Table 4 children-10-00482-t004:** Characteristics of additional included publications discussing a perinatal palliative care program.

Reference	Country	Study Design	Targeted Population	Aims/Research Questions
Akyempon et al. [34]	UK	Guideline based on recommendations and review of literature	Perinatal	Development neonatal and perinatal palliative care pathway
Bennett et al. [35]	US	Case study	Perinatal	Example of a perinatal palliative care interdisciplinary approach
Bétrémieux et al. [36]	France	Expert opinion on application of a new palliative care law	Perinatal	Application of a new palliative care law in medical practice
Breeze et al. [37]	UK	Case descriptions and comparisons	Prenatal	Examples of pregnancy termination or perinatal palliative care approach in case of lethal fetal abnormality
Cabareta et al. [38]	France	Retrospective chart review	Neonatal	Study postnatal management decisions after new neonatal palliative care protocol
Carter et al. [39]	US	Delphi method, development of a comfort care guideline using naturalistic inquiry	Neonatal	Development of a neonatal comfort care guideline
Catlin et al. [11]	US	Delphi method, build a consensus document on palliative care	Neonatal	Create a protocol for neonatal palliative care
Chamberlain et al. [40]	US	Case description	Perinatal	Case description of a perinatal assessment team
Chapman, B. [41]	New Zealand	Case study	Perinatal	Describe an approach integrating primary, secondary and community services to perinatal palliative care
Cole, J. [42]	US	Program description and case study	Perinatal	Describe a perinatal palliative care program
Conway-Orgel et al. [43]	US	Guideline development	Neonatal	Development of an algorithm and program for neonatal palliative care.
Czynski et al. [44]	US	Program description	Perinatal	Describe a perinatal palliative care program
Dahò, M. [45]	US	Semi-structured interviews	Perinatal	Study experiences of the providers working in a Perinatal Hospice program
De Burlet, A. S. [46]	Belgium	Program description	Neonatal	Describe a neonatal palliative care protocol
De Lisle-Porter et al. [47]	US	Guideline description	Neonatal	Describe an end-of-life care guideline for neonates
Denney-Koelsch et al. [4]	US	Survey	Perinatal	Survey of structure and process of perinatal palliative care programs
Engelder et al. [48]	US	Program description	Perinatal	Describe a Perinatal Comfort Care program
English et al. [49]	US	Program description	Prenatal	Describe a prenatal palliative care program
Falke et al. [50]	US	Program description	Perinatal	Describe the implementation of a perinatal hospice program
Friedman et al. [51]	US	Case study	Perinatal	Case description of a palliative care team
Gale et al. [52]	US	Program description	Neonatal	Description of a neonatal palliative care program
Garten et al. 2013 [53]	Germany	Program description	Neonatal	Recommendations for a basic neonatal palliative care concept
Garten et al. 2020 [54]	Germany	Program recommendations	Perinatal	Provide practical guidance in perinatal palliative care
Guimaraes et al. [55]	Portugal	Program description	Perinatal	Describe a perinatal palliative care program
Haxel et al. [56]	US	Retrospective descriptive consecutive case series	Neonatal	Describe outcomes of fetuses and neonates with severe or complicated congenital heart disease treated with neonatal palliative care
Hoeldtke et al. [57]	US	Program description	Perinatal	Describe a perinatal palliative care program
Jackson et al. [58]	UK	Guideline description	Perinatal	Provide guidelines for perinatal palliative care
Kauffman et al. [59]	US	Program description	Perinatal	Describe a perinatal palliative care program, including the nursing perspective.
Kenner et al. [60]	US	Program recommendations	Perinatal	Provide recommendations for the provision of family-centered perinatal palliative care
Kobler et al. [61]	US	Program description	Perinatal	Describe a perinatal palliative care service in an institution that already has a perinatal bereavement program
Korzeniewska-Eksterowicz et al. [62]	Poland	Case description	Perinatal	Describe case of the first child being cared for by the perinatal hospice
Leong Marc-Aurele et al. [20]	US	Exploratory retrospective electronic chart review	Perinatal	Examine cases referred to a perinatal palliative care team
Leuthner 1 [63]	US	Program recommendations	Neonatal	Provide recommendations for general pediatricians on neonatal palliative care
Leuthner 2 [64]	US	Reflection on fetal palliative care	Perinatal	Describe whether the fetus is a patient to whom palliative care principles apply, and which patients can receive PPC
Leuthner et al. [65]	US	Program description	Perinatal	To review perinatal palliative care concepts and describe the perinatal palliative care program
Locatelli et al. [66]	Italy	Guideline description	Perinatal	Define guidelines to establish a new hospital policy for perinatal palliative care
Longmore [67]	New Zealand	Guideline description	Neonatal	Discuss new guidelines on neonatal palliative care
Martín-Ancela et al. [68]	Spain	Guideline description	Perinatal	Describe guidelines for Perinatal Palliative Care
Miller et al. [69]	US	Program recommendations	Perinatal	Provide recommendations for perinatal palliative comfort care
Moore et al. [70]	US	Program description	Perinatal	Describe a perinatal palliative care program
Munson et al. [71]	US	Program description	Neonatal	Framework for engaging families in perinatal palliative care discussions
National association of neonatal nurses board of directors [72]	US	Guideline description (position statement)	Neonatal	Position statement of the national association of neonatal nurses on neonatal palliative care
Nolte-Buchholtz et al. [73]	Germany	Program description	Perinatal	Describe a perinatal palliative care program
Paize [74]	UK	Program description	Perinatal	Describe a perinatal palliative care program and children’s hospices
Ramer-Chrastek et al. [75]	US	Case study	Perinatal	Describe a perinatal hospice program
Roush et al. [76]	US	Program description	Perinatal	Description of a perinatal palliative care program
Rusalen et al. 2021 [77]	Italy	Literature review to propose shared care pathway	Perinatal	Propose a dedicated perinatal palliative care pathway
Rusalen [78]	Italy	Recommendation	Perinatal	Routinely employ a bioethical framework in end-of-life decision making
Ryan et al. [79]	US	Case study	Neonatal	Compare example with and without perinatal palliative care
Sidgwick et al. [9]	UK	Program description and case example	Perinatal	Describe a model for perinatal palliative care
Stringer et al. [80]	US	Guideline description	Perinatal	Provide information on perinatal palliative care
Thibeau et al. [81]	US	Reflection/literature overview	Neonatal	Standardize palliative care for both infants and their families
Tucker et al. [82]	US	Retrospective chart review	Perinatal	Describe received care in a PPC program, patient outcomes and attitudes of neonatologists
Uthaya et al. [83]	UK	Program description	Neonatal	Development of guideline for infants receiving palliative care
Wool et al. 2016 [8]	US	Cross sectional survey	Neonatal	Survey to describe existing PPC programs
Wool et al. 2020 [84]	US	Program description	Neonatal	Describe a PPC care program: history and recommendations
Ziegler et al. [85]	US	Program description	Perinatal	Describe development and processes of a PPC care program

**Table 5 children-10-00482-t005:** Overview of all involved (care) components in PPC programs.

Components	Subcomponents and Description	Times Mentioned
**Practical organization of a perinatal palliative care team**		
**PPC team members**	Core team members: - Physicians: OB-GYN, neonatologist - Nurses and midwives - Psychosocial support providers: social worker, psychologist - Care coordinator: someone of the core team specifically assigned to the family to guide them through the entire process from diagnosis until death/stillbirth and bereavement. A familiar face. Can be a physician, nurse, midwife, social services, family physician, etc., to ensure continuity of care.	N: 47 [4,8,9,11,14,15,18,19,20,26,27,28,29,30,31,33,34,35,37,39,40,41,42,43,44,48,52,53,54,56,59,62,65,68,69,71,73,74,75,76,77,78,80,81,82,84,85]
	Members included when needed: - Organ specialists: ex. pediatric cardiologist, geneticist, pediatric oncologist, etc. - Religious/spiritual support providers: chaplain, pastoral care, spiritual counselor, etc. - General practitioner: involve or inform the general practitioner - Other: NICU unit director, lactation specialist, translation services, speech pathologist, ethics committee, mediators, etc.	N: 18 [9,11,18,26,27,28,29,33,34,35,42,44,54,56,68,69,84,85]
**Activation of the perinatal palliative care team**	Which families and/or infants are eligible? Several programs discussed the inclusion criteria for contacting/starting up the perinatal palliative care program, including severe prenatal diagnosis, (extremely) premature infants, and neonates with severe conditions diagnosed after birth.	N: 24 [11,26,27,28,29,30,33,34,35,36,39,42,44,50,53,57,60,68,69,72,75,76,77,82]
	Provide perinatal palliative care irrespective of the decision made: some programs indicated their involvement in case of a severe prenatal and/or neonatal diagnosis irrespective of whether curative treatment was still sought, or palliative options were being considered.	N: 11 [4,40,57,61,65,68,72,73,74,75,77]
**Provide place and/or circumstances for palliative care**	- Ensure private time for the family - Peace and quiet - No fixed visiting hours for parents or visiting restrictions for family members and/or siblings. - Let parents/family members dictate what they do and do not need. - Homey environment. - Example: signal grieving rooms to other healthcare providers and/or parents at the ward so that others at the ward can adjust accordingly and show the appropriate amount of respect for the grieving family.	N: 31 [8,11,18,34,35,39,43,44,47,48,51,52,53,54,55,56,62,63,64,65,66,69,71,72,74,76,77,81,83,84,85]
**A step-by-step plan to organize palliative care**	Provide a checklist of steps to take in caring for families in the PPC program. Including a fixed moment to start up PPC.	N: 21 [4,11,14,15,28,30,31,33,35,39,44,46,48,52,56,66,68,74,76,80,83]
**Organization of communication between the care team**	Provide a fixed way to share information with all involved, for example by using the patients electronic form or a separate shared document. Provide a clear and detailed way of documenting all conversations and decisions made; for example by using a fixed filing system or documentation system to record provided care.	N: 40 [4,8,9,11,14,15,26,27,28,29,31,32,33,34,35,37,39,41,42,44,46,49,50,52,53,54,58,59,66,67,68,69,71,73,74,75,76,77,81,85]
	Ensure regular and multiple multidisciplinary meetings with the entire care team, to keep everyone involved up to date. Ensure that all information is passed so that parents do not have to repeat information multiple times (continuity of care).	N: 21 [9,11,14,26,29,30,41,43,44,54,59,66,67,69,71,77,78,81,82,83,85]
**Teaching of staff members**	Organize formal training, for example on: - (Perinatal) Palliative Care training - Communication training of staff members - Self-care strategies in dealing with perinatal palliative care - Training on ways to procure organs - Training on cultural and religious needs and customs - Organize a journal club to discuss new developments in perinatal palliative care.	N: 31 [4,8,11,14,15,30,31,33,34,36,38,39,44,46,47,48,50,52,53,54,55,59,60,61,66,67,68,78,81,84,85]
	Organize on the job training, for example by: - Organizing regular mortality and morbidity meetings accessible to the entire perinatal team to discuss difficult cases and/or to inform others of how a specific case was handled. - Providing palliative care orientation for new staff members. - Having the palliative care team educate/supervise/instruct other staff members on a regular basis.	N: 8 [11,31,36,39,46,52,54,84]
**Set up a collaboration with hospice or palliative home-care services**	- Transfer to home care if possible. - Provide practical support for families when going home, for example by helping them acquire the necessary medical equipment. - Possibility to transfer back to the hospital in critical situations. - Support the family at home.	N: 34 [4,8,9,11,26,28,33,34,35,36,38,41,43,44,48,49,52,54,56,58,59,61,63,64,69,71,72,73,74,76,77,83,84,85]
**A regular audit of the perinatal palliative care approach**	Several programs mentioned a regular revision of the perinatal palliative care pathway to ensure that the team can work as efficiently as possible, and new insights are being incorporated into the palliative care approach.	N: 2 [11,34]
**Fundraising for the program**	Several programs mentioned the need for perinatal palliative care to be freely available to all families regardless of the costs accompanied by providing care. Therefore, fundraising for the program was sometimes discussed as being part of the perinatal palliative care program.	N: 4 [4,31,54,76]
**Community awareness and involvement**	Offer correct knowledge on palliative care, inform the public on how they can aid in supporting families going through perinatal loss.	N: 3 [60,70,78]
**Care components of the perinatal palliative care program**		
**Child-directed care**		
**Care of the child preceding death**	Ensuring comfort of the child (non-medical) - No painful unnecessary assessments and treatments: limit uncomfortable and/or painful assessments and treatments when they are not beneficial for the wellbeing of the child. - Provide an optimal stress-free environment for the child: warmth, minimal light/noise, etc. - Holding child: skin to skin contact with parents can increase comfort of the child. When parents are not available, staff members can take over. - Stop mechanical ventilation when no longer in best interest of the child, as mechanical ventilation does not increase comfort. - Co-bedding multiples to increase comfort: letting twins/triplets/multiples share beds occasionally or permanently to increase their comfort and allow them to share skin to skin contact.	N: 40 [4,8,11,14,18,19,26,27,28,34,36,38,39,41,43,45,46,47,48,49,52,53,54,55,56,58,63,64,66,68,69,72,73,74,76,77,80,83,84,85]
	Adequate pain/symptom management - Provide comfort medication to adequately relieve pain, suffering and symptoms - Regular use of validated pain and comfort scales to assess pain and/or suffering	N: 46 [4,8,11,14,15,18,19,26,28,32,34,36,38,39,41,43,45,47,48,49,52,53,54,55,56,58,59,61,63,64,65,66,68,69,71,72,73,74,76,77,80,81,82,83,84,85]
	Food/nutrition and hydration: evaluate the need for oral, parenteral and enteral nutrition. There is discussion on whether artificial food and nutrition should continue to be provided. Breastfeeding is always considered as beneficial when the child is able to.	N: 27 [8,11,18,27,28,32,34,39,48,52,54,55,56,58,61,63,64,65,66,68,69,72,73,76,77,83,84]
	Other care provided to the child - Dressing the child - Bathing - Monitor vital signs: is not always necessary when death is expected, however monitoring vital signs can provide comfort for family members and/or healthcare providers.	N: 41 [4,8,11,15,18,26,27,28,31,32,34,36,38,39,41,42,45,46,47,48,52,53,54,55,56,59,62,63,64,65,66,68,69,72,73,76,77,80,83,84,85]
**Care of the child after death**	Care of the body: bring body to the morgue or mortuary, provide visiting options for the parents/family after death, options to cool and rewarm the body.	N: 20 [4,11,28,31,38,39,42,43,47,50,52,53,63,64,65,68,74,76,77,83]
**Post-mortem medical procedures**	- Post-mortem tests and autopsy when needed. Confirm the diagnosis after death/stillbirth. - Organ and tissue donation when possible	N: 25 [4,8,9,11,31,32,34,39,41,47,48,52,55,58,60,63,64,65,68,71,74,76,77,83,85]
**Family-directed care**		
**Maternal care**	Delivery care: ensure maternal comfort, pain management and support. Healthcare providers present during the birth to support the mother and to confirm the diagnosis.	N: 13 [4,26,28,34,42,48,53,55,59,61,65,71,75]
	Post-partum care: provide adequate post-partum care, including appropriate duration of hospital stay, providing care and pain relief, providing a lactation consultant, offer the possibility of donating breast milk, etc.	N: 16 [4,11,28,32,34,35,40,44,48,52,59,61,72,79,81,83]
**Family-centered care**	Adapt care to the values, hopes, needs and cultural background of the family. Provide very individually tailored care. This includes providing a translator when needed.	N: 46 [4,8,9,11,26,27,28,29,31,33,34,35,41,42,43,44,45,46,47,48,49,52,53,54,56,59,60,61,63,64,65,66,67,68,69,70,71,73,74,75,77,80,81,82,84,85]
**Promote family bonding and parenting**	Allow for family bonding: - Offer memory making options during life and after death of the infant. For example; extra (3D) ultrasounds, taking photos, hand and footprints, locks of hair, etc. - Organize family visits so that everyone has a chance to meet the infant. - Attempt to achieve last wishes of the family, for example going outside with their child, baptism, seeing siblings, etc.	N: 58 [4,8,11,18,20,26,27,28,29,31,33,34,35,36,38,39,41,42,43,44,45,46,47,48,49,50,51,52,53,54,55,56,57,58,59,60,61,62,63,64,65,66,67,68,69,70,71,72,74,75,76,77,79,80,81,83,84,85]
	Promote parenting: - Encourage parents to participate in caring for the child: bathing, dressing, diapering, feeding, holding, talking, etc. - Involve family members, siblings and other members of their social network.	N: 39 [4,8,11,18,27,28,31,34,35,36,38,39,42,44,45,46,47,48,52,53,54,55,58,61,62,63,64,65,66,68,69,71,72,74,76,77,79,80,85]
	Respect families’ wishes not to bond. Provide encouragement to participate in care and bonding, but do not push family members who are not willing to bond.	N: 3 [38,52,53]
**Family-centered psychosocial support**	Constant availability of psychosocial support - Provide resources for coping: social services, community resources, support between grieving parents, etc. - Support in organization: domestic help, household, etc. - Provide sibling support ⇨ Provide age-appropriate handles to inform siblings ⇨ Include siblings in care for the infant ⇨ Provide age-appropriate ways of coping - Support for the family during a new pregnancy - Support for the other parent (father, co-mother, other)	N: 61 [4,8,11,14,19,26,27,28,29,31,32,33,34,35,36,37,38,39,40,41,42,43,44,45,46,48,49,50,51,52,53,54,55,56,57,58,59,60,61,62,63,64,65,67,68,69,70,71,72,73,74,75,76,77,79,80,81,82,83,84,85]
	Bereavement support: - Provide regular and scheduled follow-up appointments through visits, calls and e-mails - Schedule follow-up appointment with physicians to address questions - Provide aid in dealing with grief: grief counseling, bereavement services, support groups in the area - Commemorate the infant: yearly bereavement ceremonies for all infants who past during that year, remembrance cards, etc.	N: 43 [4,8,11,26,28,29,31,32,33,34,35,39,40,41,42,44,46,48,51,52,53,54,55,57,59,60,63,64,65,68,69,70,71,72,74,75,76,77,79,81,83,84,85]
	Religious and/or spiritual support: assess the need for spiritual support and cultural rituals/traditions. Allow for rituals and ceremonies.	N: 49 [4,8,11,14,26,27,28,29,31,32,33,34,35,38,39,41,42,43,44,46,47,48,49,52,53,54,55,58,59,60,61,62,63,64,65,66,67,68,69,71,72,75,76,77,79,80,83,84,85]
**Practical support**	Aid in funeral planning and registering birth and death of the child. Inform, or aid parents in informing other authorities and organizations: family services, birth registries, etc., so parents do not have to make the call.	N: 33 [4,8,11,19,29,31,32,38,41,43,44,45,46,48,52,53,54,58,59,60,63,64,65,68,69,70,71,74,75,76,81,83,85]
**Healthcare provider directed care**		
**Support for healthcare providers**	Formal support for healthcare providers: Provide psychological support during working hours, for example by a psychologist, ethics consultant, psychiatry staff, social workers, etc.	N: 15 [11,20,31,33,51,52,53,59,60,61,68,69,72,81,83]
	Informal support for healthcare providers - Someone with experience present to ask questions to - Being able to indicate when healthcare providers are up to providing perinatal palliative care, and when other staff members might need to take over	N: 8 [11,52,54,67,68,79,80,83]
**Debriefings after death**	Time to reflect on what happened and address possible conflicts, problems and issues.	N: 13 [11,33,44,52,53,54,55,59,67,72,77,80,83]
**Relieve PPC members of other tasks when caring for an infant in their final moments**	Make sure that the infant and the family is cared for by fixed staff members (nurses, physicians, etc.) who are available at all times during the dying process. This is often achieved by relieving them of caring for multiple other patients during this time.	N: 5 [31,38,44,53,80]
**Advance care planning**		
**Care plan during pregnancy and/or life of the child**	Prenatal care plan - Discuss examinations needed during pregnancy - Discuss the need for private birth/labor classes - Discuss maternal health, both psychosocial and medical during the pregnancy - Discuss possibilities of non-aggressive obstetric management, neonatal palliative care options and (late) termination of pregnancy with or without feticide after 22 weeks	N: 23 [4,28,29,32,34,42,48,49,54,55,56,59,60,61,68,69,71,74,75,77,78,84,85]
	Birth plan - Discuss mode and time of delivery - Discuss the location, people present and/or spiritual support - Discuss options for maternal comfort and pain management - Discuss if fetal monitoring is required and what actions need to be taken when the child is born	N: 44 [4,9,18,20,26,28,29,31,33,34,35,37,40,41,42,44,48,49,50,53,54,55,56,57,58,59,60,61,62,63,64,66,68,69,70,71,72,73,74,75,76,77,82,85]
	Neonatal care plan - Discuss which actions are needed or if palliative care will be provided immediately after birth - Confirm the diagnosis after birth if possible - Discuss courses of action depending on the situation of the infant at birth: resuscitation, treatments, limiting care, etc.	N: 36 [4,8,9,18,20,26,28,33,34,37,38,41,42,44,49,54,56,58,60,61,63,64,65,67,68,69,71,72,73,74,76,77,78,82,84,85]
**Death plan**	- Provide parents/family members with possible scenarios of what could happen during the dying phase or stillbirth - Inquire about preferences: religious ceremonies, washing and clothing the infant, taking pictures, etc. - Bereavement care plan: anticipate and prepare for what happens after death, what support do parents anticipate needing?	N: 13 [4,11,20,31,33,34,52,63,68,72,76,77,83]
**Regular revisions of the care plan**	Option to reassess/change the plan at any time. Needed when time passes, when new diagnostic and/or prognostic information becomes available, when the medical situation of the child improves or deteriorates, or when family and/or healthcare providers change their minds.	N: 12 [9,18,26,34,37,39,44,49,54,69,74,76]
**Components regarding the decision-making process with parents/family members**		
**Which information is shared?**	Information regarding diagnosis and prognosis: be honest and open about the expected outcomes and what this means in terms of survival and morbidity. Be honest about possible diagnostic and/or prognostic uncertainty.	N: 49 [4,8,9,11,14,18,26,28,29,31,32,33,34,35,37,38,41,42,43,44,47,48,49,50,52,53,54,55,56,57,58,59,60,61,63,64,65,66,68,69,70,71,74,76,77,79,81,83,84,85]
	Prepare parents for the death, educate them on what to expect (for example, gasping)	N: 24 [4,11,28,34,36,37,38,39,41,47,48,52,53,54,55,58,59,63,64,65,70,76,77,83]
	Provide the parents with regular updates regarding the diagnosis and prognosis, or if new information becomes available.	N: 18 [8,11,26,28,29,34,38,42,43,44,47,52,53,54,69,76,81,84]
	Raise topics for discussion in family meetings when parents leave them open	N: 2 [52,54]
	Provide all possible treatment options and consequences so that parents can make an informed decision.	N: 25 [4,9,11,14,34,37,40,41,42,47,49,57,59,61,64,65,68,69,71,72,73,74,75,76,77]
**Manner of sharing information**	Schedule regular, planned, formal consultations with parents. Include all relevant members of the multidisciplinary team in order to adequately inform parents. Make sure someone is (almost) always available for questions.	N: 54 [4,8,9,11,14,15,19,20,26,27,28,29,31,32,33,34,38,40,41,42,43,44,47,48,49,52,53,54,55,56,59,60,61,62,63,64,65,66,67,68,69,70,71,72,73,75,76,77,78,80,81,82,83,84]
	Provide understandable information tailored to the parents: - Mirror terminology used by parents - Adjust to knowledge level of parents - Anticipate repeating information, as parents often miss crucial information during periods of stress - Give parents time to process information given - Involve translators when needed - Allow family members to bring others to consultations when needed - Provide reliable reading material when parents ask for it	N: 30 [8,9,26,33,34,35,38,41,42,43,44,47,49,52,53,54,55,61,63,64,66,67,69,71,73,76,77,79,83,84]
	Communication tailored to the parents’ needs, values and wishes: - Respect values and beliefs of parents - Respect parents right not to know - Provide information in a culturally appropriate manner	N: 31 [8,9,11,26,29,31,33,34,35,38,43,44,47,48,49,52,53,54,63,64,66,67,68,69,70,71,74,77,81,83,84]
	Make sure a private and comfortable room is provided for conversations with parents.	N: 15 [11,34,39,41,47,48,52,53,54,55,67,69,76,81,83]
**Shared decision making**	Make decisions together with parents. Inquire about the role parents would like to play in decision making and respect them not willing to participate in decision making if this is the case. Support patient (parent) advocacy and empower parents to participate in decision making.	N: 18 [8,9,11,28,34,35,38,39,47,49,53,54,55,68,69,74,77,79]
**Conflict resolution**	When conflict between parents and healthcare providers occurs, aim to resolve this conflict by: - Providing ethics consult when needed - Providing (cultural) mediation when needed - Referring parents to a second opinion - Preparing parents for possible differences in opinion in advance	N: 7 [11,34,53,60,69,81,83]
	Conflict resolution between healthcare providers in the team. - By being able to step down from a case - By providing mediators. - Debriefings afterwards	N: 6 [9,11,53,77,81,83]
**Care for externals**		
**Care for other families at the ward**	- Prepare other families at the ward for the death of a fellow patient. Death of a child at the ward can take away hope/make them fear for the future of their own infant. - Make sure parents are aware of the situation to avoid awkward conversations or inappropriate reactions to the grieving family.	N: 1 [53]

## Data Availability

Search strings and additional information on the search strategy are available upon written request to the corresponding author (laure.dombrecht@vub.be).

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
