# Peer review of "Components of Perinatal Palliative Care: An Integrative Review"

_children, 2023, doi:10.3390/children10030482_

Round 1

Reviewer 1 Report

this is an important review aboutperinatal palliative care; highly relevant because nowadays, many institutions and countries are beginning to develop protocols and teams dedicated to perinatal palliative care; the authors have done a good and solid job in systematically reviewing the existing literature; the method is sound (although systematic reviews are not my area of expertise), and the results and conclusison/discussion is well decsribed and usefull for clinicians working with seriously ill newborns or fetusses,. I especially found the recommendatins about fiture reserach important, i wish the authirs would have expemnded that part a bit more  I missed only one element in the analysis, and maybe this was because it wasn't reported in the reviewed papers: the costs (or cost/benefit) related to perinatal pall care teams. If more knowledge about costs is available, please spend a few lines on that subject because I believe such info could play a very important role in some institutions, in taking away some of the barriers that prevent perinatal palliative care initiatives

Reviewer 2 Report

Thank you for the opportunity to read your work. It is not possible to provide complete editorial comments on this manuscript due to the following:

1. Tables 3 and 4 are not available to review. Please make available

2. Under the results section, 3.2 is missing. Please make available

3. A body of evidence pertinent to this review is missing (quantitative articles examining quality indicators and parental satisfaction of perinatal palliative care by Wool and colleagues & qualitative articles examining parental regret by Cote-Arsenault and colleagues and another by Wool and colleagues). Consider evaluating and including.

4. The WHO definition of palliative care is incomplete. Please include full definition.

I look forward to carefully reading and responding to an updated version. Thank you.

Reviewer 3 Report

Dear Authors. Congratulations on the research. I note a number of changes. First, re-specify the objective because it is not clear. I understand it but the reading is not easy. Also, please elaborate more on the prism protocol in the methodology. On the other hand, I miss that other reviews or meta-analysis on the subject.

Round 2

Reviewer 2 Report

This paper reveals a great deal of time and effort on the part of the authors who seek to identify and report select components of program-driven perinatal palliative care. This work is appreciated. Unfortunately, this manuscript cannot be published due to the following methodological shortcomings:

1.      There is not a research question upon which the search strategy was built.

2.      According to Martinic and colleagues1, the Cochrane handbook states: A systematic review attempts to collate all empirical evidence that fits the pre-specified eligibility criteria in order to answer a specific research question. It uses explicit, systematic methods that are selected with a view to minimising bias, thus providing more reliable findings from which conclusions can be drawn and decisions made. The key characteristics of a systematic review are: a clearly stated set of objectives with pre-defined eligibility criteria for the studies; an explicit, reproducible methodology; a systematic search that attempts to identify all the studies that would meet the eligibility criteria; an assessment of the validity of the findings of the included studies, for example through the assessment of the risk of bias; and a systematic presentation, and synthesis, of the characteristics and findings of the included studies” 

If the authors are interested in revising this document to meet the above criteria (1) formulating a research question, 2) removing non-research based articles, and 3) using an evidence hierarchy to assess each article), then the results and discussion will also need to be reassessed and revised according to the findings from the quality appraisals.

If the authors opt to keep the non-research based articles, then this article needs to be relabeled as an integrative review. The 70 articles need to be measured using a standardized tool to complete a critical appraisal of each article. Including a sample size and population would strengthen the tables.

Here are some additional comments should the authors wish to revise:

Tables 3 and 4: the citation numbers on the tables do not match the manuscript reference pages (Lines 613-716). Rather Tables 3 and 4 have their own, separate reference pages. When revised, please work with the publisher to ensure Tables 3 & 4 are included in the manuscript and that the Tables have their reference page  clearly published along with it.

If authors opt to redefine this as an integrative review, there are some areas of confusion that when addressed, will prepare this article for publication, as follows:

1.      The 70 articles that were extracted for this review are not all research-based, and this fact needs to be clarified throughout the paper. It appears almost one-third of the retrieved articles are not research-based studies. Here are some select examples of the areas that need to be revised:

a.      In the PRISMA figure, it states the n = 70 ‘studies included in review’. Please modify to state ‘articles included in review’

b.     Section “2.3.1 study designs” and “2.4 “study selection”  imply only studies are included in this review. Only when the reader gets to lines 104/105 do we see non-research articles are included in this review. In Line 155 these are referred to as ‘protocol papers’. Please change the headings to “article designs” and  “article selection” respectively.

c.      The fact that this review includes papers that are not research-based is not clearly reflected in the abstract, and needs to be clarified in the abstract. (lines 14/15/16) Please revise

d.     In lines 109/110 it states “all publications that concerned palliative care for…..were included.” This statement is not in agreement with other portions of the paper, not “all publications….were included”, rather a sampling of publications that seek to meet your inclusion criteria. Please clarify and revise

Please provide consistency and clarification within the body of the paper to make clear to the reader that this review includes articles that are not research-oriented.

2.      Definition of terms about diagnosis varies throughout the paper, with use of many different terms  ‘severe’, ‘lethal’, ‘serious’, ‘life-limiting’, etc. – a variation in terms adds to confusion in this paper.

Recommend defining and using one term throughout the paper. The current literature on PPC generally uses the term ‘life-limiting.’

3.      Page 14 of this linked WHO referenced document https://apps.who.int/iris/bitstream/handle/10665/250584/9789241565417-eng.pdf?sequence=1 states palliative care neither hastens or postpones death. This portion of the WHO definition is important to families seeking care for their fetus/newborns, and to providers who deliver PPC. The entire literature base (since 1997) of PPC stems from support for parents who continue pregnancy. PPC was expressly developed for women who opt to continue a pregnancy.

There are ethical concerns in this paper wherein the authors, (lines 63/64 and 539/540), state a desire to expand the definition of PPC to families opting for termination. This content contradicts the original and essential features of the discipline of PPC which has been endorsed by the American College of Obstetricians and Gynecologist (ACOG)2 and the American Academy of Pediatrics (AAP)3. These organizations, as well as a robust and growing scientific foundation, set the state of the science that candidates for PPC are those who opt to continue a pregnancy. Further, the literature demonstrates that maternal psychoemotional experiences differ among those who terminate and those who continue the pregnancy. Maternal physical experiences and interactions with the health care team also differ. These dichotomies deserve careful study and consideration (and most likely, uniquely tailored support services based on the pathway women choose).  Clinical and scientific work, along with publishing a body of evidence, needs to be generated to assess the needs of families opting for termination and how providers can offer evidence-based support to these families.

The recommendation is to correctly define PPC as services rendered to parents who continue pregnancy, as per the current state of the science.

Line 332 and Figure 2: Consider changing the term “child-directed care” to “child-focused” care. This is because the child is not ‘directing’ their care per se, rather receiving care based on the direction of parents and providers, which is “child-centered” or “child-focused”.

Citation(s) for bullet points prior to the Discussion section would enable the reader to reference and cross-reference articles in a more thorough fashion. Consider citing areas that are more ethically complex (such as hydration/nutrition provision, etc.).

Lines 563-564: the statement “consider it pointless and possibly even tedious”  – please expand on this. Pointless and tedious to whom? The providers, parents, child?

Lines 565-566: the statement on cooling and rewarming is a bit unclear. Please separate ‘cooling and rewarming’ from organ procurement. The drivers behind these interventions and decisions may not be related and may confuse the reader since they are bundled into one sentence here.

Thank you for the opportunity to review your article. It is a real challenge to synthesize and report on components of PPC given that they vary so much from organization to organization (as evidenced in your results). Your work points to many areas of opportunities available to clinicians and researchers within a field that promises a robust future for growth.

1.      Krnic Martinic, M., Pieper, D., Glatt, A. et al. Definition of a systematic review used in overviews of systematic reviews, meta-epidemiological studies and textbooks. BMC Med Res Methodol 19, 203 (2019). https://doi.org/10.1186/s12874-019-0855-0

2.      Perinatal Palliative Care. Committee Opinion No. 786. American College of Obstetricians and Gynecologists Committee on Obstetric Practice, Committee on Ethics. Obstet Gynecol. (2019) 134:e84–9. doi: 10.1097/AOG.0000000000003425

3.      Perinatal Palliative care. Pediatrics December 2019, 144 (6) e20193146; DOI: https://doi.org/10.1542/peds.2019-3146
